# Towards Security Joint Trust and Game Theory for Maximizing Utility: Challenges and Countermeasures

**DOI:** 10.3390/s20010221

**Published:** 2019-12-30

**Authors:** Libingyi Huang, Guoqing Jia, Weidong Fang, Wei Chen, Wuxiong Zhang

**Affiliations:** 1College of Physics and Electronic Information Engineering, Qinghai University for Nationalities, Xining 810007, China; 2014006@qhmu.edu.cn (L.H.); guoqing.jia@qhmu.edu.cn (G.J.); 2Shanghai Research Center for Wireless Communication, Shanghai 201899, China; wuxiong.zhang@mail.sim.ac.cn; 3Key Laboratory of Wireless Sensor Network & Communication, Shanghai Institute of Micro-System and Information Technology, Chinese Academy of Sciences, Shanghai 201800, China; 4School of Computer Science and Technology, China University of Mining and Technology, Xuzhou 221116, China; chenw@cumt.edu.cn; 5State Key Laboratory of Synthetical Automation for Process Industries, Northeastern University, Shenyang 110819, China

**Keywords:** game theory, trust, security, equilibrium, utility

## Abstract

The widespread application of networks is providing a better platform for the development of society and technology. With the expansion of the scope of network applications, many issues need to be solved. Among them, the maximization of utility and the improvement of security have attracted much attention. Many existing attacks mean the network faces security challenges. The concept of trust should be considered to address these security issues. Meanwhile, the utility of the network, including efficiency, profit, welfare, etc., are concerns that should be maximized. Over the past decade, the concepts of game and trust have been introduced to various types of networks. However, there is a lack of research effort on several key points in distributed networks, which are critical to the information transmission of distributed networks, such as expelling malicious nodes quickly and accurately and finding equilibrium between energy assumption and high transmission rate. The purpose of this paper is to give a holistic overview of existing research on trust and game theory in networks. We analyzed that network utility can be maximized in terms of effectiveness, profits, and security. Moreover, a possible research agenda is proposed to promote the application and development of game theory and trust for improving security and maximizing utility.

## 1. Introduction

Our demand for network performance is growing. In common networks, such as distributed networks, social networks, and centralized networks, they usually have several common goals: Maximum efficiency, profit maximization, and network security improvement. There are often some risks in the normal information transmission or interaction process. For instance, malicious attacks or deceit will reduce the security of the network and even destroy the availability of the network. In this scenario, it is of utmost importance to improve the reliability of the communication link and maximize the utility of the network. In the current big data environment, the application of sensor-cloud has higher requirements for maximizing utility [1]. In recent years, some researchers have tried to improve network reliability by establishing trust models, schemes, or mechanisms, and have achieved some research results [2].

Trust concept has been widely applied in the field of network technology. Different researchers have different definitions and understandings of trust based on the different purposes and perspectives. Trust is a very broad concept, which is connected with the concepts of reliance, security, and accountability. Therefore, trust is introduced into the networks, which provides new ideas to solve the problems of network security and reliability. The application of trust in networks usually appears in the form of trust models and mechanisms. By calculating the trust value, the reliability of the network is evaluated. After determining the reliability of the network, how to choose a more efficient transmission path or information sharing party is also crucial. As a result, the application of decision mechanism has been expanded.

As a decision-making theory, game theory has been introduced in networks. The game takes place under the constraints of certain game rules, based on direct interaction of environmental conditions. Each player chooses their own strategies/actions based on the information they have in order to maximize their own profit. Game theory, as a mathematical modeling method to help rational decision makers choose conflict or cooperation in order to find an optimal behavior strategy, is widely used in various fields [3], such as biology, computer science, military strategy, sociology, etc.

There are some basic elements in the game, including player, strategy, payoff, outcome, and equilibrium. The basic assumption of game theory is that the decision-making subject is rational. Rationality refers to the strategy that players choose to maximize their returns, in order to maximize their payoff functions. The classification of game has different classification according to different benchmarks. According to whether a binding agreement is reached between the players, the game can be divided into cooperative game and non-cooperative game. It can be divided into static game and dynamic game according to whether the actions of the players are in order. In a static game, the latter does not know what strategy the first player has adopted. In a dynamic game, the latter can observe the choices made by the first player. According to the players’ full understanding of the game strategy and other players, the game is divided into complete information game and incomplete information game. In a complete information game, each player has precise information about other players, such as the strategy space and payoff function. In an incomplete information game, at least one player does not have accurate enough information about the other players. Different types of games can be used to represent different game models.

In order to have better utility, researchers have introduced the concepts of trust and game in different application scenario. By investigating the literature over the past 10 years, we have learned some of the current research status of game theory and trust. Through analysis, the application scenarios are roughly classified. In this literature, the author introduces game models into network/system to achieve specific goals. These goals can be roughly divided into the following categories: (1) Improve the network security. The improvements of network security include data protection, prevention attacks, and eliminating of malicious nodes. Game theory is used to establish an attack defense system to prevent attacks [4]. Game and trust are applied to the network at the same time, and related mechanisms are established to eliminate malicious nodes to protect the security of data [5]. (2) Maximizing utility, such as improving social efficiency and maximizing the profits of players. In the power network, the profits of suppliers and customers are often improved through games. In some networks, players’ profits are maximized through resource allocation [6]. It is also used to establish a routing mechanism or incentive to select strategies through games and improve the performance of the network [7,8,9]. Especially in the process of data transmission, the selection of sensors or nodes has a good application prospect [10,11].

In the network, not all players to which game theory is applied are perfectly rational. Thus, evolutionary game theory has been proposed. In terms of methodology, it differs from game theory’s emphasis on static equilibrium, which emphasizes a dynamic equilibrium [12]. In the application of games, in order to achieve equilibrium, players usually face the choice of competition or cooperation. Similarly, in systems with agents, competitiveness strongly increases cooperation [13]. Cooperation in the network is based on trust. As the application scenarios of games and trust, networks can be divided into various forms. Typical networks are distributed networks, centralized networks, and social networks. In a distributed network, the number of nodes is large, the structure is complex, and the connections are diverse. Centralized networks are prone to network congestion. The relationships of players in social networks are complex and dynamic. All three networks can be called complex networks. The study of complex network models has a certain guiding role in dealing with the problems existing in real networks. The structure of a system and the structure of a network are closely related to its functions. Each of the three networks covered in this paper has its own topology. The change of the topology structure affects the utility and security in the network. The relationship between trust, game theory, and networks is shown in Figure 1.

Social network refers to a relatively stable relationship system formed by the interaction between individual members of society. Social networks represent a variety of social relationships. Trust and trustworthiness are very important in social and human systems, especially when considering human and economic decisions. Therefore, the design of the game model in these studies is based on the trust relationship of the players in application scenarios [14,15,16,17,18,19].

Distributed networks are interconnected by nodes at multiple terminal distributed in different locations. Reliability is a key characteristic of distributed networks and a key factor in ensuring network availability [20]. In distributed networks, networks are often subject to malicious attacks. These attacks can disrupt the transmission link of the network, which in turn disrupts network availability. How to choose a reliable node for transmission, and how to isolate malicious nodes out from the network quickly and accurately, are crucial [21,22,23]. To find the optimal choice in distributed network, game theory is an effective solution [24,25,26]. Combining a trust relationship between nodes and games can solve problems such as path selection in distributed networks.

A centralized network is a large central system. In a centralized network, all data are stored in a central system and the terminal is the client [27]. Its main advantage is that it is convenient for centralized management, but it also causes information congestion. In centralized networks, we often see the concept of agents. During the centralized management or acquisition of information, trust between agents and agents or agents and clients is especially important. Similarly, combining game theory and trust can well maintain normal interaction in centralized network [28].

The improvement of network utility is the improvement of various performance metrics in the network. Only when the various performance metrics are improved simultaneously can the utility of the network be maximized [29]. As a tool for strategy selection and conflict resolution, game theory can solve the problem of simultaneous improvement of network performance metrics well. At the same time, the improvement of network security is also crucial. Game theory and trust can be used to evaluate the trust of network players, thereby helping the network resist security issues such as malicious attacks [30,31]. Therefore, we believe that the introduction of game theory and trust can maximize the utility of the network.

In this paper, we offer a review of the research literature based on the type of game, trust, and application scenarios, studying their applications in maximizing utility and security. The rest of the paper is organized as follow: In Section 2, we review related works. In Section 3, the literature is classified by the type of game, and analyzed and compared. In Section 4, the literature is categorized based on trust-based applications. In Section 5, the application scenarios of game and trust are classified and summarized. In Section 6, the possible future research directions are proposed. In Section 7, this paper is summarized.

## 2. Relate Summarized Works

In this section, we briefly describe our research work from three aspects: The application of game theory, the application of trust, and the application of joint game theory and trust; and give some examples in the literature. Through the investigation of literature, we have concluded that most researchers use the following types of games: Dynamic game [32,33,34,35,36,37,38,39,40,41,42,43,44,45], static game [46], non-cooperative game [47,48,49,50,51], cooperative game [52,53,54,55,56,57,58,59], security game [60,61,62,63,64], and trust game [65,66,67,68,69].

Researchers using the dynamic game have established some mechanisms or solutions for application scenarios, solving the issues of maximizing utility in the networks/systems. The dynamic game also has some applications in solving problems related to network security [35], gaining a higher network lifetime in wireless sensor network, and obtaining higher secure caching ratio of heterogeneous networks [38,42], such as improving the bandwidth utilization, network throughput, and other performance [32,34,36]. In some application scenarios, the main purpose of the game is to make an optimal pricing strategy to achieve higher utility of the system [39,40]. In some application scenarios, players choose to cooperate and ultimately achieve a win–win situation. In device-to-device (D2D) communication, cooperative game theory can promote effective cooperation between devices and improve overall performance [53,56]. Players in non-cooperative games are in a competitive relationship and are very common in many applications. In mobile cloud computing, a non-cooperative game model is established based on the competition behavior of multiple clients to maximize mobile terminal energy efficiency [50]. The key factor in the security game and the trust game is trust [66]. In some cases, only when the players trust each other can they get the best results. The purpose of the security game is to improve the security of the system and promote information sharing [60,61,62,63,64]. In addition to the explicit type of game model, researchers have established some game models based on different application scenarios [70,71,72,73,74]. Most of these game models are established with the following objectives as the ultimate goal: To establish trust relationships and increase the credibility of data, to defend against malicious attacks in the network, to help the system by choosing a better strategy, and to maximize network utility by achieving these goals [75,76,77,78,79,80,81].

From the perspective of game theory, we can get the above types. We can also analyze and summarize the research literature from the perspective of trust and application scenarios. In this literature, researchers use trust relationships, trustworthiness, and other factors for decision-making. Based on trust, the researchers proposed some trust models [34,40,49,52,53,58,59,64,69,75,79], trust mechanisms [32,42,43,78,82], and trust frameworks [57]. They provide a better selection scheme for players in the network/system through trust evaluation. In mobile social networks, researchers have used trust evaluation to propose a reliable secure caching scheme, which saves cache time and improves secure caching ratio [49,51]. In terms of network security, researchers have established intrusion detection systems through games and trust management to improve the security network [47].

The concept of trust and game has been introduced into various fields. By introducing a dynamic game model, some researchers use trust relationships to build a game model, improving the security of the networks. They establish a strategic trust mechanism in the internet of things and cyber-physical systems and use game theory to capture the adversarial and strategic nature of cyber-physical systems security [37,43]. In sensor networks, cooperative game and trust model are also combined, improving network security and enhancing the accuracy of data transmission [52,58,59]. Through the analysis of these literature, we can conclude that there are three main types of application scenarios: Centralized networks, distributed networks, and social networks. In these networks, specific objectives are achieved using trust and game. Trust games are mostly used in systems with agents, providing solutions for trust decision-making and trust management and establishing a trust relationship between all players to maximize the utility of all players [66,68,69]. In cloud services, the number of malicious services is reduced through cooperative game and trust evaluation, improving service performance in terms of response time and throughput [57].

To the best of our knowledge, there are no review articles that use our classification criteria. Therefore, this paper will review this literature from the perspective of game type, trust, and application scenarios. Based on the results of the investigation, the future research directions are put forward. The research direction we have proposed is helpful in improving the security and utility of sensor networks.

## 3. Game Theory in Networks

Decision-making is a behavior that is prevalent in politics, economy, technology, and daily life. It can achieve specific objectives and make decisions on future behaviors by analyzing, calculating, and judging the achievement of the target. In the literature that we have researched, researchers have introduced game theory into the network to improve network utility in various aspects. In this section, we analyze and summarize the types of game cited in this literature. We divide the types of game into seven categories, as shown in Table 1.

By introducing a dynamic game, the researchers proposed solutions for several aspects of the network/system. In terms of strategy selection, they proposed introducing the game theory into the peer-to-peer (P2P) energy trading problem in industrial Internet of things (IIoT), proposing an optimal pricing strategy using Stackelberg game for credit-based loans in [39]. In the vehicular networks, the optimal strategies of energy entities are analyzed by using the game theory [40]. In the process of strategy selection, in order to jointly improve the network utility from multiple angles, we are often looking for equilibrium. To improve the network utility, it is necessary to introduce dynamic game into the key factors in the network and calculate the equilibrium between them. In a military tactical network, in resource-restricted tactical environment, the equilibrium between group trust and system lifetime is achieved, and mission effectiveness is improved [34]. Based on game theory, a dynamic behavior monitoring scheme for evidences collection in wireless sensor networks is proposed. Equilibrium between network security and energy conservation can be achieved [38]. Game theory as a strategy selection scheme has also been introduced into the intrusion response systems [29]. The game theory is used to investigate the interactions among edge computing-enabled small cell base stations (ECSBSs) and mobile users, and the Nash equilibrium is calculated to jointly maximize the utilities of ECSBSs and mobile users [42]. To improve network performance, we need to use game theory to capture the interaction between devices and the cloud and calculate Nash equilibrium [43]. In the context of big data, game theory is used to analyze the profits between doctors and patients, and thus improve the total benefit of the medical system [44].

In some networks/systems, the introduction of dynamic game promotes network security and resource sharing efficiency. In cognitive multi-hop cellular networks, a new cognitive radio bandwidth-sharing scheme is proposed based on game theory, and the bandwidth utilization is improved [32]. In the spectrum sharing scheme, using game theory to quantitatively analyze the interaction between the primary users and secondary users, it can promote spectrum sharing in a tiered manner [36]. Game theory is introduced into the network to perform modeling analysis to prevent malicious node attacks or leaky deception in the network and improve the secrecy rate in the network, thereby improving the security of the network [33,35,37,45]. In device-to-device communication, the introduction of game theory has increased the utility of operators [31]. In the smart grid in a large population regime, the introduction of game theory has maximized the profit of providers and users [18]. Contrary to dynamic game, static game is less used. In the literature [46], static game is introduced into cloud radio access network (C-RAN). Researchers analyzed whether the operators would like to collaborate with each other by applying game theory and improve the utility between operators. In [19], the problem of efficiently coordinating price-responsive appliances operating in the electricity market is tackled within a game theoretical framework.

According to whether the players cooperate, the game is divided into cooperative game and non-cooperative game. The common goal of both games can maximize network utility and improve network security. Combining the non-cooperative game model with trust management enables peers in the intrusion detection network to collaborate truthfully and actively [47]. In wireless sensor networks, the introduction of game theory improves the defense’s strategy and helps the network against malicious nodes [30]. In a network with mobile users/terminals, the introduction of a game model can improve the utility of mobile users [49,50]. In merchant transmission planning, the introduction of game theory can improve social efficiency [20,22]. In [24], a resource scheduling method for guaranteeing resource utilization and service performance is presented, maximizing the profits of players. The cooperative game is introduced into the network, which enables the network to maintain high security against malicious attacks and reduce malicious services [52,57,58,59]. In device-to-device communication, game theory is combined with social ties or social trust. It is used to promote efficient cooperation among devices in the network to improve network performance [53,54,56].

In some servers or authentication protocols, a security game is often introduced. The introduction of game theory improves security in systems or protocols [60,61,64]. In the system with information exchange, the establishment of the game model ensures the safe and reliable transmission of information [62,63].

In multi-agent systems, the trust issues of all parties are often involved. Finding a reliable agent in the system is crucial. The introduction of trust games into these systems has greatly helped the trust decisions in the system [66,68,69]. The introduction of trust game has provided dynamic decision-making for network proactive defense [6].

Some researchers have proposed specific game models for corresponding systems. Combining the game model with the trust relationship improves the reliability of the data in the network [75,76,80]. The introduction of the game model improves the security of the network and against external and internal attacks [14,70,72,74,77]. There are also some game models that provide great help for network or system strategy selection [27,71,78,79,81], or addressed the problem of resource allocation [26]. In [28], the introduction of game theory optimizes the probability of successful transmission probabilities of data and the caching probabilities in some networks.

By analyzing this literature, we can obtain the following conclusions. The introduction of game theory provides a new solution for improving network security and maximizing utility. The dynamic game model is used most widely in the use of game theory. This is because, in some networks/systems, players usually have some knowledge of the information and decisions of other players. Therefore, modeling it as a dynamic game is more appropriate. Cooperative game, non-cooperative game, and security game also occupy a certain proportion. When two or more metrics in the network are optimized for conflict, the introduction of game theory solves the problem. Researchers use equilibrium to solve multi-metric optimization conflicts, thereby maximizing network utility. In some networks, information or resources need to be shared. The introduction of game theory provides a good strategy for resource sharing. When there is a malicious attack in the network, whether it is an external attack or an internal attack, the combination of game theory and trust relationship provides a solution to solve the security problem.

In general, the introduction of game theory has maximized utility from the following aspects. (1) It improves the security and secrecy rate of the network, against malicious attacks, and reduce malicious services; (2) it maximizes the profits of players in the network, thereby maximizing network utility; and (3) it promotes information/resource sharing to provide network players with better strategy selection.

## 4. Trust Model, Trust Mechanism, and Trust Framework in Networks

The concept of trust has been widely introduced into networks. When there are multiple players in the network, trust becomes a key evaluation metric for each player’s decision-making [25]. Many scholars have introduced trust and reputation into the network, and improved network utility through trust management against various malicious attacks [65,67], improving network security. In the literature that we researched, the introduction of the concept of trust provides better performance for the network. Some researchers have evaluated key factors in the network by using trustworthiness or trust value [83,84]. Some researchers have introduced the trust model, trust mechanism, or trust framework in the network [85,86]. Here, we only classify and elaborate the trust model, trust mechanism, and trust framework, as shown in Table 2.

The trust model is introduced into the wireless sensor network, by using trust evaluation to improve the security of the network [52,58,64,87]. There are fixed relationships in some networks, such as social trust or social reciprocity. Modeling trust models based on these social relationships improves the data rate in D2D communications [53,79]. It can also improve social welfare in social networks [69]. For some networks with operators or mobile users, the introduction of the trust model provides good help for users’ choice [40,49]. In military tactical network and multi-agent systems, there are often multiple players. The introduction of the trust model promotes the trust relationship between players, thereby improving mission effectiveness [34,75]. The introduction of the trust mechanism is often to solve the problem of security and efficiency in the network [42,43]. For example, in a P2P network, the trust mechanism can help peers make interaction decisions and prevent malicious peers [82]. Trust mechanisms can also increase the efficiency of spectrum and bandwidth sharing in the network [32,78]. The information sharing and interaction in the network are carried out normally, and the overall efficiency of the network is improved [17]. In the fight against collusion attacks in the cloud service community, the introduction of a trust framework minimizes the number of malicious services [57]. In [15], researchers have proposed a dynamic trust framework to detect selfish and malicious nodes, and the entire network utility has been improved. Not only for the network, but the routing protocols also improve their security due to the introduction of trust [21].

The introduction of trust provides a solution to maximizing network utility from another perspective. Different from the introduction of game theory, the introduction of trust is more targeted. Trust is introduced in networks that have trust relationships. In these networks, trust is an important metric for information interaction [73]. The introduction of the trust model mainly solved the problems of social welfare and social efficiency. The introduction of trust mechanism and trust framework solved the problem of resource sharing and malicious attacks. Therefore, the use of trust to solve problems in the network has broad research prospects.

## 5. Analysis and Comparison Based on Application Scenarios

We have classified the scenarios of game theory and trust application. It is mainly divided into three types of application scenarios: Centralized network, distributed network, and social network. We refined the metrics involved in evaluating network utility for different application scenarios, as shown in Figure 2 and Table 3. We have summarized it in the first subsection. The second subsection discussed the differences between the application of game theory to different types of networks (i.e., social networks, centralized networks, distributed networks).

### 5.1. Application Scenarios

A centralized network facilitates data management. In a centralized network, data are processed centrally and all information is managed by a database. The terminal is only used for input and output. In our research literature, researchers combine game theory and trust to optimize networks or systems, including wireless network, cyber-physical systems, multi-agent system, and other application scenarios. In a centralized network, utility is typically evaluated by using the following metrics: Secure caching ratio and average saved time [42,49,51]; service cost, service quality, or transmission overhead [54,70,74]; social welfare [68,83]; network throughput or total amount of data transferred [32,78]; energy efficiency or operation efficiency [46,50]; or detecting the network’s against attack ability and defense costs through simulation experiments [37,43,81].

Distributed networks have high reliability. All nodes are randomly distributed in the network and there is no management of the central node. There are many path selections during information transmission, and resource sharing is easy, but it is vulnerable to malicious nodes. In a distributed network, network performance is often evaluated through security and efficiency [52,55,82,84], or metrics based on energy efficiency, trustworthiness of data transmission, and cost-efficiency [38,39,47,58,59].

In social networks, each member of the society has formed a relatively stable relationship system due to interaction. Social networks often represent a variety of social relationships. The social network can be either an individual or an organization. In social networks, network throughput and data storage rate are often used as evaluation metrics [15,57,79,85]; or energy efficiency, information transmission rate, and social welfare are used as metrics [48,53,69]. There are also some social networks that protect players’ privacy through trust evaluation [16].

In summary, different networks have different metrics. Centralized networks are prone to network congestion, so their metrics are often related to transmission efficiency or time overhead. Distributed networks are vulnerable to various attacks, so their metrics are often related to security and accuracy. Social networks are most concerned with social efficiency, therefore often assessed on the basis of throughput and social welfare. The introduction of game theory and trust provides a good solution for improving these metrics in the network.

### 5.2. Differences in Applying Game Theory in Different Networks

Different types of networks have different topologies and therefore face different problems. The three networks discussed in this paper are members of complex networks, and their network characteristics determine the different applications of game theory in networks. The characteristics of centralized data management in a centralized network cause problems such as being prone to network congestion and slow processing speed. The application of game theory in centralized networks can ensure convenient data processing in the network while improving the efficiency of the network (that is, maximizing the network utility in terms of transmission overhead and effectiveness). The combination of trust and game theory can better ensure the security and reliability of information storage. In a distributed network (such as a typical wireless sensor network), a large number of nodes are randomly distributed in the network, and the structure of the network is complex and changeable. The application of game theory and trust in distributed networks can help the network form reliable transmission links. While ensuring the reliability of the network, it also ensures the security of information transmission, resists malicious attacks, and maximizes the utility of the network. In social networks, the relationships between players are complex and dynamically changing, especially the trust relationships between players. The introduction of game theory and trust helps players in social networks to better cooperation and maximize network utility.

## 6. Future Research Direction

In the literature we researched, we analyzed that the structure of the network is basically fixed in centralized networks and social networks. Once these fixed links have problems, it is hard to solve them quickly. It may cause the network to be unavailable in a short time. Especially when the management node fails, it will affect the working efficiency of the whole network. In the case of large network traffic, it may affect data transmission efficiency. Thus, the distributed network reflects its own advantages. The main feature of the distributed network is that there is no central node in the network, and the communication control functions are distributed on each node. The reliability of the distributed network is high, and the nodes in the network share the resources conveniently, and it can also improve the information flow distribution of the line. The sensor network is the main representative of the current distributed network. However, in the literature that we researched, fewer researchers have introduced game theory and trust into sensor networks.

In the existing literature, the researchers only considered part of the network performance; for example, network security, network lifetime, and data forwarding rate [38,58]. It does not consider how to choose a reliable node for data transmission. Nor did it suggest how to expel a malicious node from the network as soon as possible to ensure the security of the network. There is also no better strategy between network energy consumption and higher transmission efficiency. In future research work, we can introduce the concepts of game theory and trust into sensor networks. According to the characteristics of the sensor network, relevant mechanisms/strategies can be formulated to improve the transmission efficiency and security of the network and to select the security node and expel malicious nodes quickly and accurately. It is also possible to find equilibrium between low power consumption and high transmission efficiency through game theory. The network security, reliability, and other metrics can be optimized.

## 7. Conclusions

This paper starts from the perspective of game theory and trust and classifies and compares the research literature. The results of research on maximizing utility and improving security were studied. The application scenarios of these game theories are analyzed. Through analysis, most researchers have cited dynamic game and non-cooperative game. Trust is also widely used in the network, trust value, and trust account for the majority. Some literature use trust for modeling to form the trust model, trust mechanism, or trust framework. The concepts of game theory and trust have been introduced into various networks in the literature of research. It can be roughly divided into social networks, centralized networks, and distributed networks. However, no researchers have applied the combination of game theory and trust to sensor networks. 

Our contribution mainly involves two aspects: (1) Through the work of Section 3, Section 4, and Section 5, the combination of trust and game theory can improve network utility from three aspects: Against malicious attacks, maximizing the profits of players, and improving resource efficiency, especially in terms of security. (2) We point out potential research directions based on the overall trends observed from the survey results. We believe that game theory as a theory to resolve conflict situations can solve some conflict situations in sensor networks: Trustworthiness and efficiency. The concept of trust can also be used as an effective evaluation metric when the network node selects the next node. Trust can also be used for network security evaluation: To discover malicious nodes and expel them from the network as soon as possible. We believe that the combination of game theory and trust can simultaneously improve the overall network utility and network security. Overcoming the shortcomings of vulnerability and energy limitation in sensor networks makes sensor networks run more efficiently. Therefore, in future research, we can introduce game theory and trust into the sensor network to maximizing utility and improving network security.

## Figures and Tables

**Figure 1 sensors-20-00221-f001:**
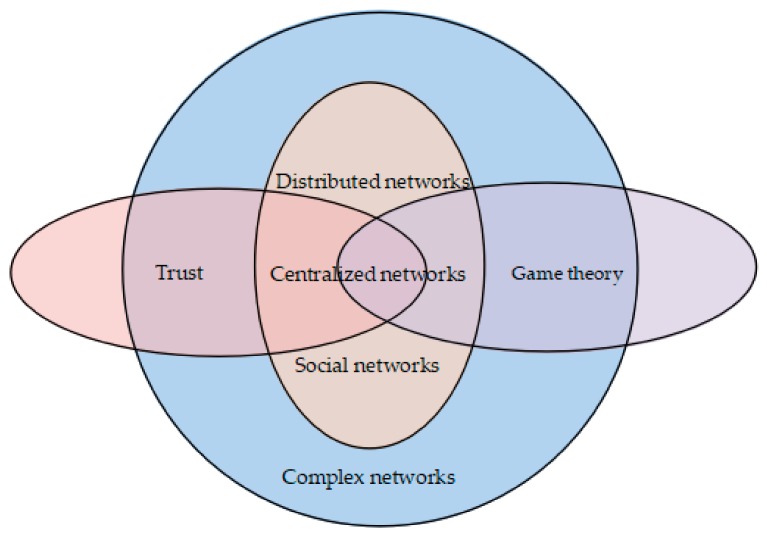
The relations between complex network, trust, and game theory.

**Figure 2 sensors-20-00221-f002:**
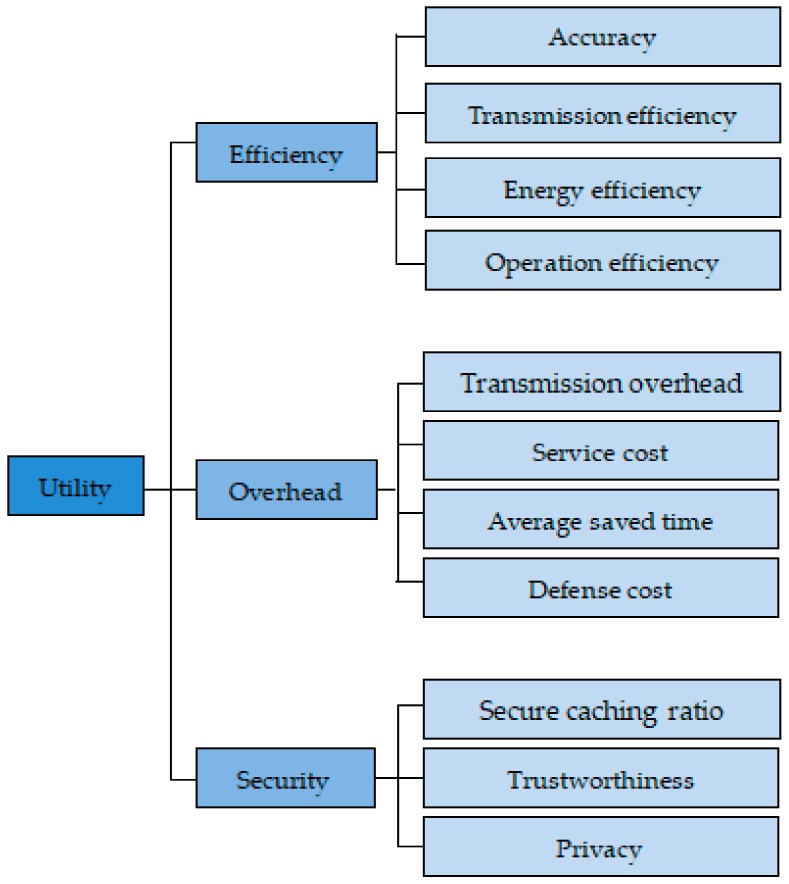
Some performance metrics related to utility.

**Table 1 sensors-20-00221-t001:** Examples of games based on their types.

**Dynamic Game**	**Static Game**	**Non-Cooperative Game**	**Cooperative Game**
Kim [32] Ho et al. [33] Cho [34] Al-Talabani et al. [35] Muthukumar and Sahai [36] Pawlick and Zhu [37] Yang et al. [38] Li et al. [39] Wang et al. [40] Gao et al. [41] Xu et al. [42] Pawlick et al. [43] Niu et al. [44] Pawlick et al. [45] Shi et al. [31] Maharjan et al. [18] Zonouz et al. [29]	Tian et al. [46] De Paola et al. [19]	Zhu et al. [47] Anand et al. [48] Xu et al. [49] Ahn et al. [50] Xu et al. [51] Borjigin et al. [24] De Paola et al. [20] Gong et al. [22] Basilico et al. [30]	Duan et al. [52] Chen et al. [53] Cao et al. [54] Cho et al. [55] Waqas et al. [56] Wahab et al. [57] Abdalzaher et al. [58] Rani et al. [59]
**Security Game**	**Trust Game**	**O** **thers**	
Wu et al. [60] Ruan et al. [61] John et al. [62] Wu et al. [63] Jin and van Dijk [64]	Yu et al. [66] Abbass et al. [68] Chica et al. [69] Hu et al. [6]	Zhu et al. [70] Ding et al. [71] Zhang et al. [72] Aloqaily et al. [74] Rishwaraj et al. [75] Pouryazdan et al. [76] Chen et al. [77] Lorenzo et al. [78] Li et al. [79] Aroyo et al. [80] Moghadam and Modares [81] Cui et al. [28] Yin et al. [26] Chen et al. [27] Li et al. [14]	

**Table 2 sensors-20-00221-t002:** Classification of trust applications.

Trust Model	Trust Mechanism	Trust Framework
Duan et al. [52] Chen et al. [53] Cho et al. [34] Rishwaraj et al. [75] Chica et al. [69] Xu et al. [49] Li et al. [79] Wang et al. [40] Abdalzaher et al. [58] Jin and Van Dijk [64] Rani et al. [59]	Wu et al. [82] Kim [32] Lorenzo et al. [78] Xu et al. [42] Pawlick et al. [43]	Wahab et al. [57] Wang et al. [15]

**Table 3 sensors-20-00221-t003:** Classification of application scenarios.

Distributed Network	Centralized Network	Social Network
Wu et al. [82] Zhu et al. [47] Ho et al. [33] Duan et al. [52] Cho et al. [56] Yang et al. [38] Li et al. [39] Gao et al. [41] Abdalzaher et al. [58] Wang et al. [86] Kang et al. [84] Rani et al. [59]	Kim [32] Yu et al. [66] Zhu et al. [70] Aloqaily et al. [74] Cao et al. [54] Abbass et al. [68] Rishwaraj et al. [75] Pawlick and Zhu [37] Xu et al. [49] Ahn et al. [50] Lorenzo et al. [78] Xu et al. [51] Tian et al. [46] Xu et al. [83] Xu et al. [42] Pawlick et al. [43] Moghadam and Modares [81]	Anand et al. [48] Chen et al. [53] Chica et al. [69] Zhang et al. [85] Li et al. [79] Wahab et al. [57] Wang et al. [15] Yan et al. [16]

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
