# Peer review of "Towards Security Joint Trust and Game Theory for Maximizing Utility: Challenges and Countermeasures"

_sensors, 2019, doi:10.3390/s20010221_

Round 1

Reviewer 1 Report

In this work, authors provide a general overview on networked technological systems, focusing in particular on the relation between these structures and game theory.
Introduction provides a general description of the context, emphasising on the aspects that might deserve more attention (e.g. efficiency, security etc), moving to the definition of “trust”. Then, game theory models are presented. Section Related Works provides a general view on the existing literature on the topic. The next sections as Game Theory in Networks cover the single aspects of this manuscript, ending with Future Direction and Conclusions.

In general, this manuscript can be of interest for readership of this journal. However, before to provide a final recommendation, the following points should be considered:

(1) Given the focus on Game Theory and Networks, there is a literature that should be at least mentioned due to its relevance in the area of social networks, complex networks and related dynamics, i.e. evolutionary games. Here, some references that authors can consider (covering the game aspects, that of networked systems, and that of social dynamics) are: “The Role of competitiveness in the Prisoner’s Dilemma, Computational Social Networks, 2(15), 2015”; “Statistical Physics and Computational Methods for Evolutionary Game Theory, Springer, 2018”

(2) Adding few Figures, to highlight relations between main concepts here presented, would be helpful and I suppose also more effective than tables.

(3) Also considering the previous references, I suggest to introduce a short section on Network theory (complex networks theory). Notably, authors do massive utilisation of the related concepts without to introduce them. For instance, how and why different topologies are able to influence social dynamics on networks? Questions like that have been addressed by a number of investigations, and the outcomes are relevant for the topic of this manuscript. I think that a brief section on this topic would be useful and would also make the manuscript stronger.

(4) Authors might consider to improve Conclusions, that in the current form seems more a very brief summary concepts previously reported than a section aimed at presenting relevant information. For instance, they could discuss more the differences between the game theoretic aspects when applied to the various classes of networks (i.e. social networks, centralised networks, distributed networks). In addition, the combination between Trust and Games is a further aspect to discuss in more detail.

Author Response

Comments and Suggestions for Authors

In this work, authors provide a general overview on networked technological systems, focusing in particular on the relation between these structures and game theory. Introduction provides a general description of the context, emphasising on the aspects that might deserve more attention (e.g. efficiency, security etc), moving to the definition of “trust”. Then, game theory models are presented. Section Related Works provides a general view on the existing literature on the topic. The next sections as Game Theory in Networks cover the single aspects of this manuscript, ending with Future Direction and Conclusions.

In general, this manuscript can be of interest for readership of this journal. However, before to provide a final recommendation, the following points should be considered:

(1) Given the focus on Game Theory and Networks, there is a literature that should be at least mentioned due to its relevance in the area of social networks, complex networks and related dynamics, i.e. evolutionary games. Here, some references that authors can consider (covering the game aspects, that of networked systems, and that of social dynamics) are: “The Role of competitiveness in the Prisoner’s Dilemma, Computational Social Networks, 2(15), 2015”; “Statistical Physics and Computational Methods for Evolutionary Game Theory, Springer, 2018”

Re: many thanks, we have accepted your suggestion and added relevant content about complex network in the Introduction, and cited the literature you recommended. (Page 2)

(2) Adding few Figures, to highlight relations between main concepts here presented, would be helpful and I suppose also more effective than tables.

Re: thank you very much for your comments, and we have added figures based on the relevant content in Sections 1 and Section 5.

(3) Also considering the previous references, I suggest to introduce a short section on Network theory (complex networks theory). Notably, authors do massive utilisation of the related concepts without to introduce them. For instance, how and why different topologies are able to influence social dynamics on networks? Questions like that have been addressed by a number of investigations, and the outcomes are relevant for the topic of this manuscript. I think that a brief section on this topic would be useful and would also make the manuscript stronger.

Re: many thanks, we have added content about the network in the Introduction, mentioning the impact of topology on network security and utility. We also added a subsection in Section 5, which mentioned that the utility of network enhancement is different for different structures.

(4) Authors might consider to improve Conclusions, that in the current form seems more a very brief summary concepts previously reported than a section aimed at presenting relevant information. For instance, they could discuss more the differences between the game theoretic aspects when applied to the various classes of networks (i.e. social networks, centralized networks, distributed networks). In addition, the combination between Trust and Games is a further aspect to discuss in more detail.

Re: thank you very much for your suggestions, we have added a subsection in section 5, that outlines the differences in applying game theory to different networks. We have improved the Conclusion section to highlight our contributions.

Reviewer 2 Report

This work deals with trust and game theory in networks. Authors provide information, recorded in the literature and categorize the published works based on trust classification and application scenarios. The findings of this work are interesting for this research area, but authors’ purposes are not highlighted, and it is difficult to understand their contribution and its importance. It seems that their contribution is the categorizations, described in Sections 4 and 5, but they should explain it more.

Additionally, the paper needs improvements in some parts. Specifically:

In general, more references should be provided after statements, e.g. line 147 “The key…trust”. Some paragraphs should be combined, due to their length, such as paragraph 5 and paragraph 7 of the Introduction. In the Abstract section, the connection between sentences should be revised, e.g. the last two sentences. In the Introduction, the selection of game theory should be justified more. In Section 2, it would be better to explain the acronym “D2D”. Similarly, in Section 4 regarding “P2P”. The information of Section 2 and 3 are similar. It seems that Section 3 is the analysis of Section 2, as the types of games are described. Authors should highlight the differences between these two sections, otherwise, they could be combined. The title of the first table is not absolutely connected with its context. Please, revise the title, e.g. probably, “Examples of games based on their types”.

Concluding, the work needs improvement in terms of each language (grammar syntax errors) in some parts e.g. line 379, the “1.”, probably should be deleted or line 219, “In [22], the researcher….”.

Author Response

Comments and Suggestions for Authors

This work deals with trust and game theory in networks. Authors provide information, recorded in the literature and categorize the published works based on trust classification and application scenarios. The findings of this work are interesting for this research area, but authors’ purposes are not highlighted, and it is difficult to understand their contribution and its importance. It seems that their contribution is the categorizations, described in Sections 4 and 5, but they should explain it more.

Additionally, the paper needs improvements in some parts. Specifically:

In general, more references should be provided after statements, e.g. line 147 “The key…trust”.

Re: many thanks, we have considered putting reference [64] here as support.

Some paragraphs should be combined, due to their length, such as paragraph 5 and paragraph 7 of the Introduction.

Re: many thanks, the fifth and sixth paragraphs have been combined and the content of the seventh paragraph was reconstructed.

In the Abstract section, the connection between sentences should be revised, e.g. the last two sentences.

Re: many thanks, we have modified the connection between sentences.

In the Introduction, the selection of game theory should be justified more.

Re: many thanks, we have added a paragraph to explain more about the selection of game theory.

In Section 2, it would be better to explain the acronym “D2D”. Similarly, in Section 4 regarding “P2P”.

Re: many thanks, we have explained acronyms.

The information of Section 2 and 3 are similar. It seems that Section 3 is the analysis of Section 2, as the types of games are described. Authors should highlight the differences between these two sections, otherwise, they could be combined.

Re: many thanks, we have restructured the content of Section 2, which summarized our investigating from three aspects: trust, game theory, and joint trust and game theory. The Section 3 is a detailed description of the application of game theory in networks.

The title of the first table is not absolutely connected with its context. Please, revise the title, e.g. probably, “Examples of games based on their types”.

Re: thank you very much for your suggestions, we have revised the title.

Concluding, the work needs improvement in terms of each language (grammar syntax errors) in some parts e.g. line 379, the “1.”, probably should be deleted or line 219, “In [22], the researcher….”.

Re: many thanks, we have corrected some syntax errors.

Round 2

Reviewer 1 Report

Authors improved their manuscript. I can now recommend it